# Heterogeneous Federated PEFT via Sparse Mixture-of-Experts

## Abstract

Parameter-efficient fine-tuning (PEFT) has emerged as a critical technique for adapting large language models (LLMs) in federated learning (FL), enabling resource-efficient model updates without compromising user privacy. However, existing FL approaches predominantly rely on a single PEFT type shared across all clients, limiting their ability to handle the substantial data heterogeneity. In this work, we propose **Hermes**, a novel federated PEFT (FedPEFT) framework that introduces the concept of Heterogeneous FedPEFT, where each client flexibly combines multiple PEFT (*e.g.*, LoRA, Adapter, Prefix-tuning) to better fit local data distributions. To address key challenges such as gradient conflicts, expert underutilization, and biased aggregation arising from this heterogeneous design, Hermes employs a structured sparse mixture-of-experts architecture with gradient-aware gating, loss-free bias adjustment, and inverse-frequency aggregation strategies. These techniques jointly ensure stable optimization and balanced contribution across clients. Extensive experiments on multiple NLP benchmarks demonstrate that Hermes achieves superior personalization performance compared to state-of-the-art homogeneous FedPEFT baselines, highlighting its potential as an effective solution for federated LLM fine-tuning under non-IID settings.

## 1 Introduction

Recently, large language models (LLMs), such as GPT Cong-Lem et al. (2025), PaLM Chowdhery et al. (2023) and LLaMA Grattafiori et al. (2024), have attracted significant attention across a wide range of domains. As LLMs become increasingly prevalent, adapting them to specific downstream tasks has become essential. For example, LLMs can be customized to analyze local medical data collected by different institutions in the medical scenario Chen et al. (2024c). However, the limited availability of labeled data presents a major obstacle to the effective utilization of LLMs in such scenarios. Moreover, compliance with privacy regulations (*e.g.*, GDPR Protection (2018)) prohibits institutions from sharing sensitive data, further restricting centralized training. Federated learning (FL) Chen et al. (2024b); Li et al. (2020) has emerged as a powerful paradigm that allows distributed model training in numerous clients without exchanging sensitive local data. Despite this advantage, applying LLMs in FL introduces substantial challenges.

Modern LLMs contain billions of parameters, making full fine-tuning in FL prohibitively expensive in terms of computation and communication overhead Kuang et al. (2024). To efficiently adapt LLMs in FL, PEFT techniques, including LoRA Hu et al. (2022), Adapter Houlsby et al. (2019), and Prefix-tuning Li & Liang (2021), have emerged as practical tools. By updating only a small subset of parameters while keeping the backbone model frozen, PEFT significantly reduces the resource demands of LLM adaptation. This has led to a paradigm known as FedLLM Sun et al. (2024a), which integrates PEFT into FL to enable efficient federated LLM fine-tuning. However, one of the key challenges of FedPEFT is data heterogeneity Cho et al. (2024); Wang et al. (2024), where the distribution of the data or the characteristics of the tasks are not identified and independent (*non-IID*). Therefore, achieving strong personalization for each client is particularly difficult.

In the literature, existing work adopts a homogeneous FedPEFT paradigm, where all clients use the same type of PEFT throughout federated training, as shown in Fig. 1(a). For example, Bian et al. (2025); Peng et al. (2025) incorporate local fine-tuning of the global PEFT or introduce regularization between global and local PEFT. To further mitigate data heterogeneity, Su et al. (2024); Singhal

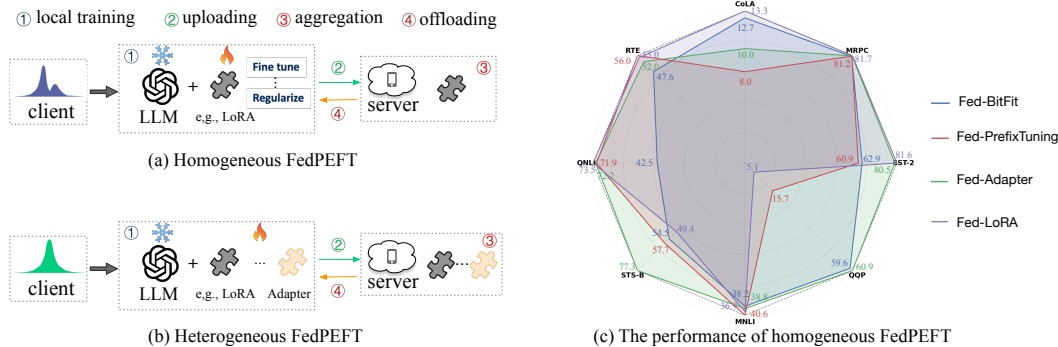

Figure 1: The architecture of (a) homogeneous FedPEFT using the same PEFT (e.g., LoRA) and (b) heterogeneous FedPEFT combining multiple PEFT (e.g., LoRA, Adapter).(c) Accuracy (%) across eight tasks, showing that no single PEFT consistently excels on all datasets.

et al. (2025); Liu et al. (2025) design dual or decoupled PEFT into shared and private components, which maintain separate components for global knowledge aggregation and local personalization, effectively reducing interference between generalized and client-specific representations. Tran et al. (2025); Cui et al. (2024); Xie et al. (2024); Long et al. (2024) adopt a model mixture strategy for personalization, where clients maintain a global PEFT and a local PEFT simultaneously. However, these approaches fundamentally rely on a single PEFT type per client, inherently limiting personalization in non-IID federated environments. We simulate four FedPEFT to reveal this limitation, as illustrated in Fig. 1(c). None of the methods performs consistently well across all datasets. For instance, while Fed-LoRA achieves strong accuracy on SST-2 and MRPC, it underperforms on CoLA and QQP. These findings highlight that enforcing the same PEFT architecture across heterogeneous clients leads to suboptimal performance, as no single PEFT is universally optimal in an FL setting.

In contrast, we propose a promising paradigm, which we term heterogeneous FedPEFT, as illustrated in Fig. 1(b). Rather than relying on a single, predetermined adaptation strategy, Heterogeneous Fed-PEFT allows each client to combine multiple PEFT (e.g., LoRA, Adapter, and Prefix-tuning) to better capture local data distribution. Theoretically, integrating multiple PEFT into a heterogeneous FedPEFT framework could provide stronger personalization by selecting the most suitable PEFT modules for each individual client or task. However, realizing this theoretical potential is non-trivial. Naively combining multiple PEFT in federated learning introduces several issues. *First*, gradient conflicts frequently arise, as different PEFT often optimize in inconsistent directions. This interference destabilizes training and slows convergence. *Second*, utilization imbalance is common, since clients naturally prefer certain PEFT aligned with their data, leaving others underused and preventing effective learning. *Third*, federated aggregation becomes heterogeneous and biased, because clients contribute updates from different subsets of PEFT with varying scales and frequencies, complicating server aggregation. These issues highlight that simple heterogeneous multiple PEFT without principled coordination is insufficient and may even degrade performance compared to homogeneous baselines.

To address the above challenges, we propose **Hermes**, a novel FedPEFT framework that unifies multiple PEFT through a structured sparse mixture-of-experts design. Specifically, to mitigate gradient conflicts across different PEFT, we design a sparse MoE architecture where PEFT are decoupled into independent experts, and a sparse top-$k$ gating mechanism selectively activates only a subset of them. This expert independence and sparsity ensure that each PEFT contributes updates from distinct subspaces, effectively mitigating conflicts and improving training stability. To address PEFT utilization imbalance, we propose a local updating balance strategy combining gradient-aware gating and loss-free dynamic bias adjustment. The gradient-aware gating mechanism jointly considers token representations and gradient alignment signals to guide expert selection more effectively, while the dynamic bias adjustment adaptively recalibrates gating probabilities based on historical activation frequencies to avoid utilization skew. Finally, to aggregate heterogeneous updates across clients, we propose a global aggregation balance strategy that rescales each PEFT's updates inversely proportional to their average activation rate before applying parameter averaging. This pre-

vents frequently selected experts from dominating the aggregated parameters and ensures balanced contributions from all PEFT. The contributions of this paper are summarized as follows.

- We introduce the concept of Heterogeneous FedPEFT, which enables clients to flexibly combine multiple PEFT within federated LLM fine-tuning. We formally prove that our framework achieves a better loss bound.

- We propose Hermes, a novel FedPEFT framework that integrates a sparse MoE architecture with gradient-aware gating, loss-free dynamic bias adjustment, and inverse-frequency aggregation to effectively coordinate heterogeneous PEFT.

- We conduct evaluations on benchmark datasets, demonstrating that Hermes consistently achieves SOTA performance, surpassing existing FedPEFT.

## 2 RELATED WORK

**Parameter Efficient Fine-Tuning.** The PEFT strategies can be broadly classified into four categories. First, additive PEFT modifies the model architecture by injecting new trainable modules or parameters. For example, adapters Pfeiffer et al. (2021); Jie et al. (2024); Lei et al. (2023); Edalati et al. (2025) are inserted following the FFN layer to enhance the computational efficiency. Prefix-tuning Li & Liang (2021); Li et al. (2023); Zhang et al. (2023) introduces learnable vectors that are prepended to keys and values across all transformer layers. Second, selective PEFT makes a subset of parameters trainable during fine-tuning Fu et al. (2023); Das et al. (2023); Liao et al. (2023). Third, reparameterized PEFT constructs a reparameterization of the original model parameters for training, then equivalently transforms it back for inference, such as LoRA Hu et al. (2022); Zhang et al. (2024b); Yang et al. (2025); He et al. (2025). Finally, hybrid PEFT combines advantages from different PEFT methods to build a unified PEFT model. For instance, UniPELT Mao et al. (2022) integrates LoRA, prefix-tuning, and adapters into each transformer block. S4 Chen et al. (2023) explores design spaces for several PEFT methods to uncover underlying design patterns.

**Federated PEFT.** PEFT techniques have been integrated into FL to minimize communication costs and maximize efficiency Zhang et al. (2024a); Bai et al. (2024); Sun et al. (2024b); Wu et al. (2024a); Che et al. (2023); Xu et al. (2024). Recent works introduce personalization into federated LLM fine-tuning via personalized PEFT modules Yang et al. (2023); Yi et al. (2023); Guo et al. (2023); Sun et al. (2023), dual adapter integration Long et al. (2024); Chen et al. (2024a); Xie et al. (2024) and dual LoRA Qi et al. (2024); Hao et al. (2025). However, these approaches have limitations: many methods Yang et al. (2023); Qi et al. (2024) only support parameter heterogeneity within the same model architecture, limiting personalization. Others Long et al. (2024); Chen et al. (2024a) rely on manually defined private architectures or hyperparameters, leading to suboptimal performance. Recently, the MoE-based approach has been promising for personalized federated learning, as it personalizes models to specific data domains through expert collaboration Guo et al. (2021); Yi et al. (2024); Qiao et al. (2024). Some studies apply MoE to federated LLM fine-tuning using lightweight PEFT as experts to reduce resource consumption. Methods include mixture of prompt-based experts Luo et al. (2025), dual LoRA expert integration Wu et al. (2024b), and cluster-based LoRA expert combination Almansoori et al. (2024). However, these approaches only support parameter heterogeneity within the same model architecture.

## 3 PRELIMINARIES AND PROBLEM FORMULATION

### 3.1 PEFT APPROACHES

**Adapter.** Adapter Houlsby et al. (2019) adds a trainable bottleneck layer after the feedforward network in each Transformer layer of an LLM. Each adapter consists of a down-projection weight $W_{down} \in \mathbb{R}^{r \times d}$, an up-projection weight $W_{up} \in \mathbb{R}^{d \times r}$, and a non-linearity $\varphi(\cdot)$, such as ReLU. The adapter residual is defined as $\Delta E_{\theta_A}(h) = W_{up} \cdot \varphi(W_{down} \cdot h)$, where $h \in \mathbb{R}^d$.

**LoRA.** LoRA Hu et al. (2022) only updates the parameters of some layers, *e.g.*, self-attention. Consider a layer in the network, LoRA freezes the pre-trained weight $W_0 \in \mathbb{R}^{d_1 \times d_2}$ and inserts a trainable update in the form $\Delta E_{\theta_L}(h) = (BA)x$ where $A \in \mathbb{R}^{r \times d_2}$, $B \in \mathbb{R}^{d_1 \times r}$, and $r \leq \min(d_1, d_2)$.

To ensure consistency with the pre-trained weight during the initial phase, $B$ is initialized as a zero matrix, while $A$ is initialized with Gaussian noise $\mathcal{N}(0, \sigma^2)$.

**Prefix-tuning.** Prefix-tuning Li & Liang (2021) optimizes a small set of continuous task-specific vectors, called *prefixes*, which are prepended to the input sequence at each Transformer layer. Concretely, for a given layer, Prefix-tuning introduces trainable key-value pairs $(K_p, V_p) \in \mathbb{R}^{l_p \times d}$, where $l_p$ is the prefix length and $d$ is the hidden size. During self-attention, the output is modified as $\Delta E_{\theta_P}(h) = \text{Attn}(Q, [K; K_p], [V; V_p]) - \text{Attn}(Q, K, V)$, $[K; K_p]$ means concatenating the original keys with the *prefixes*.

### 3.2 PROBLEM FORMULATION

We consider a typical FL setting with $n$ clients, where each client $i$ holds a training dataset $\mathcal{D}_i$. Let $\Theta$ represents the frozen parameters of the pre-trained LLM, and $\theta$ denote a single PEFT module's parameters shared across all clients. $f_i(\theta) := \mathbb{E}_{\mathbf{x}_i \sim \mathcal{D}_i}[\ell(\mathbf{x}_i|\Theta, \theta)]$ is a loss evaluated on an instance $\mathbf{x_i}$ sampled from local data $\mathcal{D}_i$ of client $i$. A common homogenous objective of FL is to optimize a single global PEFT module that minimizes the weighted average loss among all clients.

$$\underset{\theta}{\text{minimize}} \sum_{i=1}^{n} \alpha_i f_i(\Theta, \theta), \tag{1}$$

where the weights $\alpha_i > 0$ satisfy $\sum_{i=1}^{n} \alpha_i = 1$. A common choice is $\alpha_i = \frac{|\mathcal{D}_i|}{\sum_{i=1}^{n} |\mathcal{D}_i|}$. However, the one-model-fits-all formulation in Eq. (1) is inadequate under heterogeneous client data, since different clients may benefit from different PEFT strategies.

To better handle heterogeneity, personalized FedPEFT allows each client to maintain multiple PEFT, with potentially different choices and priorities. Specifically, we consider three widely used modules: Adapter (A), LoRA (L), and Prefix (P). Each client $i$ learns its own set of PEFT parameters.

$$\underset{\{\theta_{m,i}\}_{i=1}^{n}}{\text{minimize}} \sum_{i=1}^{n} \alpha_i (f_i(\Theta, \theta_{m,i}), \quad m \in \{A, L, P\} \tag{2}$$

where $\theta_{A,i}, \theta_{L,i}, \theta_{P,i}$ are the parameters of Adapter, LoRA and Prefix-tuning for the client $i$.

The Eq. (2) highlights that each client is equipped with multiple PEFT experts, and the choice of which experts to activate and update can vary across clients depending on their data distributions and tasks. The overall challenge is to design routing and aggregation mechanisms that support such heterogeneous multi-PEFT personalization in a federated environment.

## 4 METHODOLOGY

### 4.1 OVERVIEW

Fig. 2 illustrates the overall architecture of Hermes, our proposed heterogeneous FedPEFT framework. The framework consists of client-side modular fine-tuning with unified PEFT blocks and server-side global aggregation with balance control.

On the client side, each Transformer block is augmented with three PEFT that form a unified PEFT expert set. These PEFT operate in parallel on the hidden representations, generating candidate updates $E_{\Delta \theta_A(x)}, E_{\Delta \theta_L(x)}, E_{\Delta \theta_P(x)}$. A lightweight router is then employed to determine which subset of PEFT should be activated for a given input. The router integrates two complementary signals: a gradient-aware score, which highlights PEFT with stronger learning signals, and a loss-free bias adjustment, which prevents expert starvation. Based on these scores, the router selects the top-$k$ experts within each block to produce the final output via sparse aggregation. Only the parameters of the selected experts are locally updated and uploaded, ensuring communication efficiency.

On the server side, Hermes collects the uploaded top-$k$ PEFT from multiple clients. Since different clients may upload different subsets of PEFT, a naïve averaging would unfairly favor frequently selected PEFT. To address this, we design a global balance aggregation mechanism. For each expert type, the server aggregates parameters across clients with an inverse-frequency weighting scheme,

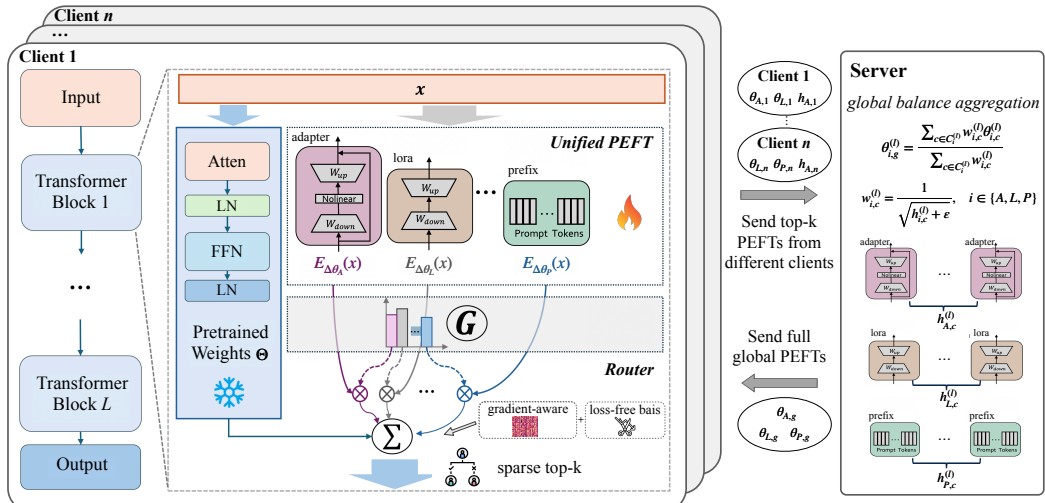

Figure 2: An overview of Hermes.

which down-weights frequently selected experts and up-weights rare but informative ones. This strategy mitigates aggregation imbalance and preserves diversity among experts. The aggregated global experts are then synchronized and broadcast to all clients, providing them with an updated and balanced pool of PEFT modules for the next training round.

## 4.2 CLIENT-SIDE MODULAR DESIGN

### 4.2.1 UNIFIED PEFT

To accommodate heterogeneous client distributions while maintaining a consistent interface for routing and aggregation, we unify multiple PEFT techniques under a common residual formulation. Consider client $c$ and Transformer layer $l$. Let $h_{\text{base}}^{(l)}$ denote the base (LLM) representation leaving layer $l$ that will be fed to layer $l+1$. We attach three PEFT to this layer, as introduced in Section 3.1. Each module independently produces a residual transformation

$$h_m'^{(l)} = h_{\text{base}}^{(l)} + \Delta E_{\theta_{m,c}}^{(l)}(h^{(l-1)}), \quad m \in \{A, L, P\}, \tag{3}$$

where $\Delta E_{\theta_{m,c}}^{(l)}$ is the expert-specific residual function parameterized by $\theta_{m,c}$. Adapters apply bottleneck MLPs in the feed-forward sublayer, LoRA injects low-rank updates into attention projections, and Prefix-tuning augments the key–value space of attention with learnable vectors.

### 4.2.2 ROUTER DESIGN

While unified PEFT provides a common residual representation for heterogeneous modules, an effective mechanism is still required to determine which experts should be activated and how their contributions should be weighted. To this end, we design a *dual-perspective router* that integrates both input semantics and learning dynamics. This router assigns non-negative weights $(\alpha_{A,c}^{(l)}, \alpha_{L,c}^{(l)}, \alpha_{P,c}^{(l)})$ to the three experts at layer $l$ of client $c$, where the weights sum to one.

**Feature view.** The first perspective comes from the hidden representation itself. Given the input $h^{(l-1)}$, the router computes a feature-based score vector via a lightweight gating network $G_f$:

$$s_{m,c}^{(f,l)} = G_f(h^{(l-1)}), \quad m \in \{A, L, P\}. \tag{4}$$

These scores capture the affinity between the input features and each expert, reflecting how suitable an expert is for the current token or batch. After normalization by a softmax function, they form the feature-view probabilities.

**Gradient view.** The second perspective relies on the optimization dynamics. We measure the learning utility of each expert by monitoring the magnitude of its gradient contribution. For client $c$

and PEFT $j$, the gradient-view score is defined as

$$s_{m,c}^{(g,l)} = \text{Norm}\Big(\big\|\nabla_{\theta_{m,c}^{(l)}} \mathcal{L}_c\big\|\Big),\tag{5}$$

where $\mathcal{L}_c$ is the local training loss. This score reflects how much improvement in loss can be achieved if the expert $m$ is updated. To reduce oscillation, we maintain an exponential moving average (EMA) of these values across training steps.

**Loss-free bias adjustment.** A common issue in expert routing is *expert starvation*, where a subset of experts monopolizes selection while others are rarely updated. To address this, we introduce a bias term $b_{m,c}^{(l)}$ that is updated outside the gradient graph according to the selection frequency:

$$b_{m,c}^{(l)} \leftarrow b_{m,c}^{(l)} + \gamma_b \cdot \frac{\bar{n}^{(l)} - n_{m,c}^{(l)}}{\bar{n}^{(l)}},\tag{6}$$

where $n_{m,c}^{(l)}$ is the number of tokens assigned to expert $m$ in the current batch, $\bar{n}^{(l)}$ is the ideal balanced count, and $\gamma_b$ is a bias learning rate. This adjustment boosts underutilized experts and discourages over-selected ones, without introducing an explicit loss penalty.

**Final routing weights.** The final score for each expert combines the two perspectives with a trade-off coefficient $\gamma$ and adds the loss-free bias:

$$r_{m,c}^{(l)} = \gamma\, s_{m,c}^{(f,l)} + (1-\gamma)\, s_{m,c}^{(g,l)} + b_{m,c}^{(l)}.\tag{7}$$

We then apply a top-$k$ operator to select the most relevant experts, setting the weights of unselected experts to zero. For the selected experts, the routing weights are normalized via a sparse softmax:

$$\alpha_{m,c}^{(l)} = \frac{\exp(r_{m,c}^{(l)})}{\sum_{m \in \mathcal{S}_c^{(l)}} \exp(r_{m,c}^{(l)})}, \quad m \in \mathcal{S}_c^{(l)},\tag{8}$$

where $\mathcal{S}_c^{(l)}$ is the set of top-$k$ experts chosen for client $c$ at layer $l$. The final hidden representation is obtained as a weighted mixture of the expert-specific residuals.

$$h^{(l)} = \sum_{m \in \mathcal{S}_c^{(l)}} \alpha_{m,c}^{(l)} \Big( h_{\text{base}}^{(l)} + \Delta E_{\theta_{m,c}}^{(l)}(h^{(l-1)}) \Big).\tag{9}$$

Here $h_{\text{base}}^{(l)}$ is the base representation, $\Delta E_{\theta_{m,c}}^{(l)}(h^{(l-1)})$ is the residual of expert $m$ defined in Section 4.2.1. Eq. (9) unifies the outputs of heterogeneous PEFT modules into a single representation $h^{(l)}$ that is passed to the next Transformer layer.

### 4.3 Server-side Aggregation

After local training and expert routing, each client uploads only the parameters of its selected top-$k$ PEFT modules. This selective communication substantially reduces bandwidth usage but also introduces heterogeneity across clients, as different subsets of experts are uploaded in each round.

**Expert aggregation.** We treat each expert type at each Transformer layer as an independent aggregation unit. Let $\mathcal{C}_m^{(l)}$ as the set of clients that uploaded expert $m$ at this layer, the server aggregates these parameters as

$$\theta_{m,\text{g}}^{(l)} = \frac{1}{Z_m^{(l)}} \sum_{c \in \mathcal{C}_m^{(l)}} \omega_{m,c}^{(l)}\, \theta_{m,c}^{(l)}, \qquad Z_m^{(l)} = \sum_{c \in \mathcal{C}_m^{(l)}} \omega_{m,c}^{(l)},\tag{10}$$

where $\omega_{m,c}^{(l)}$ is a weight assigned to client $c$'s contribution.

**Inverse-frequency weighting.** For the aggregation weights, a naive choice is to set $\omega_{m,c}^{(l)} = 1$, which reduces the scheme to FedAvg over the selected clients. However, such uniform averaging is problematic in our setting: experts that are selected by many clients dominate the updates, while rarely selected but potentially valuable experts may be marginalized. To address this, we propose an *inverse-frequency weighting* scheme.

$$\omega_{m,c}^{(l)} = \frac{1}{\sqrt{h_{m,c}^{(l)} + \varepsilon}},\tag{11}$$

---

**Algorithm 1:** Hermes

---

1: **Server initialization.** Initialize all PEFT parameters $\{\theta_{m,g}^{(l)}\}$ for $m \in \{A, L, P\}, l = 1, \ldots, L$ and broadcast to clients.

2: # **Server Aggregation**

3: For each layer $l$ and expert $m$, collect updates from clients $\mathcal{C}_m^{(l)}$.

4: Aggregate global expert $\theta_{m,g}^{(l)}$ using weighted average Eq. (10), with inverse-frequency scaling Eq. (11).

5: Broadcast all global experts $\{\theta_{m,g}^{(l)}\}_{m \in \{A,L,P\}}$ to every client.

6: # **Client Update**

7: **for** each communication round $t = 1, 2, \ldots, T$ **do**

8:    **for** each client $c \in \{1, \ldots, n\}$ **in parallel do**

9:       Receive global experts $\{\theta_{m,g}^{(l)}\}_{m \in \{A,L,P\}}$ from server.

10:      For each layer $l$, compute unified residuals $h_m^{(l)}$ via Eq. (3).

11:      Compute feature-view scores $s_{m,c}^{(f,l)}$ and gradient-view scores $s_{m,c}^{(g,l)}$ via Eq. (4)–(5).

12:      Update bias $b_{m,c}^{(l)}$ according to Eq. (6).

13:      Fuse into routing logits $r_{m,c}^{(l)}$ (Eq. (7)), apply top-$k$ selection, and normalize to obtain weights $\alpha_{m,c}^{(l)}$ (Eq. (8)).

14:      Aggregate selected experts to produce $h^{(l)}$ via Eq. (9).

15:      **Local update.** Optimize parameters $\theta_{m,c}^{(l)}$ of selected experts using local data.

16:      Upload $\{\theta_{m,c}^{(l)}\}_{m \in \mathcal{S}^{(l)}}, h_{m,c}^{(l)}$ of selected experts to server, where $h_{m,c}^{(l)}$ is usage count.

17:    **end for**

18: **end for**

---

where $h_{m,c}^{(l)}$ is the usage count of expert $m$ on client $c$, and $\varepsilon$ is a small constant for numerical stability. This scaling reduces the relative weight of overused experts while amplifying the contribution of underrepresented ones, thereby preventing mode collapse into a few dominant modules.

**Global broadcast.** Once the global parameters $\theta_{m,g}^{(l)}$ are obtained, the server broadcasts the *entire set* of experts $\{\theta_{m,g}^{(l)}\}_{m \in \{A,L,P\}}$ to all clients. This design ensures that even clients which did not upload a particular expert in the current round receive its updated version.

### 4.4 CONVERGENCE GUARANTEES

We now state the convergence result of our heterogeneous FedPEFT framework. All technical assumptions (A1–A6), supporting lemmas, and complete proofs are deferred to the Appendix. The main theorem shows that our method achieves the same $O(1/\sqrt{T})$ convergence rate as FedAvg, with additional error terms that are explicitly controlled by routing and aggregation.

**Theorem 4.1** (**Convergence to stationary points**). *Let $F(\theta)$ be a smooth, possibly non-convex objective with L-Lipschitz continuous gradients. Assume unbiased stochastic gradients with variance bounded by $\sigma^2$, and suppose the router and aggregation satisfy Assumptions A1–A6. If the learning rate $\eta$ and local step count $E_p$ are chosen such that $0 < \eta < \frac{2}{LE_p}$, then after $T$ communication rounds the averaged squared gradient norm satisfies*

$$\frac{1}{T} \sum_{t=1}^{T} \mathbb{E}\big[\|\nabla F(\bar{\theta}^t)\|^2\big] \leq \underbrace{\mathcal{O}\Big(\tfrac{F(\bar{\theta}^0) - F^\star}{\eta T E_p}\Big)}_{optimization} + \underbrace{\mathcal{O}\Big(\tfrac{\sigma^2}{n}\Big)}_{stochastic\ noise} + \underbrace{\mathcal{O}\big(\epsilon_{sel}^2 + \epsilon_{agg}^2\big)}_{router\ and\ aggregation} .$$

**Interpretation.** As the number of communication rounds $T$ grows, the optimization error decays at the standard rate for smooth non-convex FL. The additional terms $\epsilon_{\text{sel}}$ and $\epsilon_{\text{agg}}$ capture the bias from Top-$k$ routing and the variance from inverse-frequency aggregation, respectively. Both are rigorously bounded in Appendix (Lemmas 4.3 and 4.4), ensuring that with proper design of router bias and aggregation weights, these terms remain small and do not hinder convergence.

Table 1: Federated GLUE under Dirichlet splits. Top: stronger non-IID ($\alpha$=1.0); Bottom: near IID ($\alpha$=10). Metrics are $acc$ except MRPC=$F1$, CoLA=$mcc$, STS-B=$spearman$.

| Method | GLUE, RoBERTa-base | | | | | | | | GLUE, LLaMA-3.2-3B | | |
| --- | --- | --- | --- | --- | --- | --- | --- | --- | --- | --- | --- |
| | SST-2 | MRPC | CoLA | RTE | QNLI | STS-B | MNLI | QQP | SST-2 | CoLA | QQP |
| $\alpha$=1.0 (non-IID) | | | | | | | | | | | |
| FedFT | 92.5 | 89.2 | 61.5 | 68.8 | 89.5 | 88.1 | 84.9 | 88.0 | 91.0 | 56.1 | 87.2 |
| FedAdapter | 91.8 | 87.6 | 60.5 | 67.9 | 89.4 | 87.6 | 84.5 | 86.9 | 90.6 | 55.4 | 86.4 |
| FedLoRA | 92.1 | 83.0 | 59.5 | 65.9 | 89.3 | 86.5 | 83.4 | 85.9 | 89.9 | 53.8 | 85.3 |
| FedPrefix | 90.5 | 84.3 | 55.5 | 56.1 | 85.1 | 83.5 | 79.7 | 83.2 | 88.9 | 50.2 | 83.1 |
| FedUniPELT | 89.8 | 81.9 | 55.9 | 59.8 | 86.1 | 83.6 | 80.1 | 82.9 | 89.2 | 53.1 | 85.7 |
| Hermes | 93.4 | 90.1 | 63.0 | 70.4 | 90.5 | 89.2 | 86.0 | 89.1 | 92.1 | 57.7 | 88.2 |
| $\alpha$=10 (near IID) | | | | | | | | | | | |
| FedFT | 93.6 | 90.3 | 62.6 | 69.9 | 90.6 | 89.2 | 86.0 | 89.1 | 92.3 | 58.3 | 88.4 |
| FedAdapter | 93.9 | 89.7 | 62.6 | 70.0 | 91.5 | 89.7 | 86.6 | 89.0 | 92.6 | 58.5 | 88.9 |
| FedLoRA | 94.2 | 85.1 | 61.6 | 68.0 | 91.4 | 88.6 | 85.5 | 88.0 | 92.1 | 57.2 | 87.8 |
| FedPrefix | 93.8 | 87.6 | 58.8 | 59.4 | 88.4 | 86.8 | 83.0 | 86.5 | 91.6 | 55.1 | 86.7 |
| FedUniPELT | 93.3 | 88.5 | 60.9 | 67.1 | 90.6 | 88.9 | 85.0 | 87.2 | 91.8 | 56.8 | 87.1 |
| Hermes | 95.0 | 91.8 | 64.6 | 72.0 | 92.1 | 90.9 | 87.6 | 90.7 | 93.5 | 60.1 | 89.2 |

## 5 EXPERIMENT

### 5.1 SETUP

**Dataset**. We evaluate our method on the General Language Understanding Evaluation (GLUE) benchmark Wang et al. (2018), a widely adopted suite for measuring natural language understanding. The benchmark encompasses four categories of tasks, namely linguistic acceptability (CoLA), sentiment analysis (SST-2), similarity and paraphrase detection (MRPC, STS-B, QQP), and natural language inference (MNLI, QNLI, RTE).

**Non-IID partitioning.** Following Lin et al. Lin et al. (2022), we generate heterogeneity by sampling class-prior vectors from a Dirichlet distribution. Specifically, we draw $D \sim \mathrm{Dir}(\alpha)$ and allocate data $D_k$ to the $k$-th client according to $D$. The parameter $\alpha$ controls the non-IID level, and a smaller $\alpha$ yields a stronger label distribution shift. Unless otherwise noted, we set $\alpha = 1.0$ throughout.

**Baselines.** Under the FedAvg framework, we evaluate *full fine-tuning* (**FedFT**) and three representative PEFT: Adapter (Houlsby et al., 2019) (**FedAdapter**), and LoRA (Hu et al., 2022) (**FedLoRA**), Prefix-tuning (Li & Liang, 2021) (**FedPrefix**). In addition, since **UniPELT** (Mao et al., 2022) uses multiple PEFT techniques, we further implement its federated version (**FedUniPELT**) to ensure a comprehensive comparison against our method. Our proposed **Hermes** extends this line by introducing a MoE-style unified PEFT with router-based top-$k$ selection and usage-aware aggregation.

**Implementation Details.** For the backbone encoder, we adopt **RoBERTa-base** (125M parameters) for GLUE experiments and **LLaMA-3.2-3B** for large-scale ablations. Hermes unifies all three PEFT modules as parallel experts in each Transformer block, coordinated by a gradient-aware router with top-$k$ expert selection ($k = 2$ by default). Clients perform $E_p = 10$ local epochs per round, with Adam optimizer, learning rate $2 \times 10^{-4}$, batch size 32, and weight decay $10^{-4}$. Router temperature $\tau = 0.1$ and bias step size $\gamma_b = 0.05$ are used for loss-free balancing. On the server, we apply inverse-frequency aggregation with $\varepsilon = 1$. The adapter bottleneck dimension is set to $r = 16$, the LoRA rank is $r = 8$, and the prefix length is $l_p = 16$ per layer.

### 5.2 PERFORMANCE

**Exp-1: Main results.** Table 2 summarizes the performance on GLUE under both stronger non-IID ($\alpha$=1.0) and near IID ($\alpha$=10). Hermes consistently outperforms all baselines. Single-PEFT methods exhibit task-dependent strengths but degrade substantially on harder datasets such as CoLA and RTE. FedUniPELT, while combining multiple modules, suffers from gradient interference, yielding only marginal gains. In contrast, Hermes consistently outperforms all baselines, with larger improvements under non-IID ($\alpha = 1.0$). The gains arise from its MoE-style routing and inverse-frequency aggregation, which mitigate expert underutilization and prevent domination by frequent modules. Furthermore, on LLaMA-3.2-3B, the trend remains, showing that the proposed mechanisms scale effectively to larger backbones and heterogeneous federated environments.

Table 2: The effectiveness of key components in **Hermes** (A = aggregation balance, K = top-$k$ router, G = gradient-aware gating, F = feature-view scores).

| Ablation | SST-2 | MRPC | CoLA | RTE | QNLI | STS-B | MNLI | QQP |
|---|---|---|---|---|---|---|---|---|
| $\alpha$=1.0 (non-IID, RoBERTa-base) | | | | | | | | |
| Hermes-w/o-A | 92.7 | 89.9 | 62.75 | 69.9 | 90.1 | 88.7 | 85.5 | 88.8 |
| Hermes-w/o-K | 92.7 | 89.1 | 62.27 | 69.5 | 89.9 | 88.5 | 85.4 | 88.3 |
| Hermes-w/o-G | 92.1 | 88.9 | 62.04 | 69.4 | 89.2 | 87.8 | 84.9 | 87.9 |
| Hermes-w/o-F | 92.2 | 88.9 | 61.65 | 69.5 | 85.4 | 86.3 | 85.3 | 87.7 |
| Hermes | 93.4 | 90.1 | 63.0 | 70.4 | 90.5 | 89.2 | 86.0 | 89.1 |

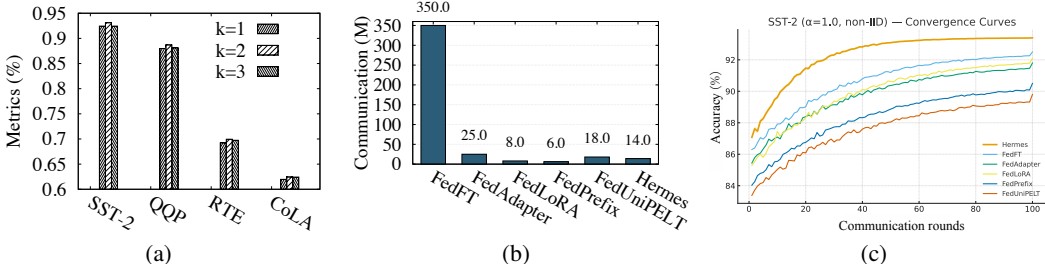

(a)  (b)  (c)

Figure 3: (a) The impact of varying the number of activated experts $k$. (b) The communication cost each round. (c) The convergence comparison on SST-2 under non-IID split ($\alpha = 1.0$).

**Exp-2: Ablation of components.** Table 2 reports ablation results under $\alpha$=1.0. Removing any key component leads to a performance drop, verifying their necessity. In particular, discarding the feature-view scores (w/o-F) produces the most severe degradation, highlighting that semantic input signals are crucial for expert selection. Similarly, removing the top-$k$ router (w/o-K) or gradient-aware gating (w/o-G) reduces effectiveness by impairing the routing quality, and omitting aggregation balance (w/o-A) introduces update dominance. The complete Hermes consistently achieves the best results, confirming the complementarity of its design.

**Exp-3: Impact of the number of activated PEFT.** We further examine the effect of varying the number of activated experts $k$ in Fig. 3(a). Activating a single expert ($k$=1) underutilizes model capacity, while selecting three experts ($k$=3) introduces redundancy without additional benefits. The best performance is observed at $k$=2, which strikes a favorable balance between expert diversity and training stability. This result indicates that sparse but non-trivial expert activation is the most effective setting for Hermes in federated PEFT.

**Exp-4: Communication analysis.** Figure 3(b) reports the communication cost per round. Hermes maintains a cost comparable to single-PEFT baselines since only the selected experts are uploaded. FedUniPELT incurs much higher overhead because it synchronizes all modules regardless of their contribution. This validates that sparse MoE-style activation not only improves accuracy but also preserves communication efficiency, which is critical in federated training.

**Exp-5: Convergence analysis** Fig. 3(c) presents convergence curves on SST-2 under the non-IID ($\alpha$=1.0). Hermes achieves both faster and smoother convergence compared to all baselines. Single-PEFT methods converge more slowly and plateau at lower accuracy due to limited adaptation capacity. FedUniPELT initially progresses quickly but soon stagnates, reflecting gradient interference between modules. In contrast, Hermes maintains steady improvement and reaches higher accuracy within fewer rounds.

## 6 CONCLUSION

We introduce Hermes, a heterogeneous FedPEFT framework that unifies multiple PEFT modules via a sparse mixture-of-experts design. By combining gradient-aware routing, loss-free bias adjustment, and inverse-frequency aggregation, Hermes achieves superior personalization performance compared to state-of-the-art homogeneous FedPEFT baselines.

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

# A  THEORETICAL ANALYSIS

We study the convergence properties of our federated hybrid PEFT framework. We first formalize the setup and assumptions, then establish stability, routing, selection bias, and aggregation variance lemmas, before presenting the main convergence theorem. Each lemma is not standalone: it contributes directly to bounding specific error terms $\epsilon_{\text{sel}}$ or $\epsilon_{\text{agg}}$ that appear in the final theorem.

## A.1  SETUP

Let the global objective be

$$F(\Theta, \theta) = \sum_{c=1}^{n} \alpha_c f_c(\Theta, \theta), \quad f_c(\Theta, \theta) = \mathbb{E}_{(x,y) \sim \mathcal{D}_c}[\ell(y \mid x; \Theta, \theta)]$$

where $\Theta$ is the frozen backbone and $\theta = \{\theta_m^{(l)}\}_{l,m}$ collects all PEFT parameters. Client $c$ holds $\mathcal{D}_c$ and updates a local copy $\theta_c$ for $E_p$ steps before uploading selected experts to the server for aggregation.

**Unified PEFT formulation**. At layer $l$, denote by $h^{(l-1)}$ the routed input and $h_{\text{base}}^{(l)}$ the frozen backbone output. For each expert $m \in \{A, L, P\}$ and client $c$, we write the residual as

$$\Delta E_{\theta_{m,c}}^{(l)}(h^{(l-1)}), \qquad h_m'^{(l)} = h_{\text{base}}^{(l)} + \Delta E_{\theta_{m,c}}^{(l)}(h^{(l-1)}).$$

The Top-$k$ router selects $\mathcal{S}_c^{(l)}$ with weights $w_{m,c}^{(l)}$ summing to one, and the layer output is

$$\tilde{h}^{(l)} = \sum_{m \in \mathcal{S}_c^{(l)}} w_{m,c}^{(l)} h_m'^{(l)}.$$

This unified residual view allows LoRA, Adapter, and Prefix-tuning to be analyzed within the same theoretical framework, ensuring consistency in later error analysis.

## A.2  ASSUMPTIONS

We adopt the following standard conditions:

**Assumption A1 (L-smoothness)**. *For each client $c$, $f_c(\Theta, \theta)$ is L-smooth in $\theta$:*
$$\|\nabla f_c(\Theta, \theta) - \nabla f_c(\Theta, \theta')\| \leq L \|\theta - \theta'\|$$

**Assumption A2 (Stochastic gradients)**. *For a minibatch $B_c$,*
$$\mathbb{E}[\hat{g}_c \mid \theta] = \nabla f_c(\Theta, \theta), \qquad \mathbb{E}\|\hat{g}_c - \nabla f_c(\Theta, \theta)\|^2 \leq \sigma^2.$$

**Assumption A3 (Expert regularity)**. *Each residual map $\Delta E_{\theta_{m,c}}^{(l)}$ is $L_E$-Lipschitz in input and $\rho_E$-Lipschitz in parameters, with Jacobian norm $\|J_m^{(l)}\| \leq \kappa_E$.*

**Assumption A4 (Router regularity & Top-$k$ sparsity)**. *The router produces $w_{m,c}^{(l)}$ via a Lipschitz function of $(\varphi(h^{(l-1)}), \bar{g}_{m,c}^{(l)}, b_{m,c}^{(l)})$ with temperature $\tau > 0$, and exactly $k$ non-zero entries per layer. Let $M^{(l)} = |\mathcal{M}^{(l)}|$ be the number of experts at layer $l$.*

**Assumption A5 (Loss-free bias updates)**. *Bias evolves as*
$$b_{m,c}^{(l)} \leftarrow b_{m,c}^{(l)} + \gamma_b^t \frac{\bar{n}^{(l)} - n_{m,c}^{(l)}(t)}{\bar{n}^{(l)}}, \quad \bar{n}^{(l)} = \frac{B \cdot k}{M^{(l)}},$$
*where $\gamma_b$ is small enough for stochastic approximation stability.*

**Assumption A6 (Inverse-frequency aggregation)**. *The server aggregates as*
$$\theta_{m,g}^{(l)} = \frac{1}{Z_m^{(l)}} \sum_{c \in \mathcal{C}_m^{(l)}} \omega_{m,c}^{(l)} \theta_{m,c}^{(l)}, \qquad \omega_{m,c}^{(l)} = (h_{m,c}^{(l)} + \varepsilon)^{-\alpha}, \qquad Z_m^{(l)} = \sum_{c \in \mathcal{C}_m^{(l)}} \omega_{m,c}^{(l)},$$

*with $\alpha \in [0,1]$ and $\varepsilon > 0$. We typically set $\alpha = \frac{1}{2}$ but keep it symbolic for analysis. Here $h_{m,c}^{(l)}$ is the token-level usage count reported by clients. Across clients, we assume the local noises are independent (or weakly correlated so that Cauchy-Schwarz yields the same functional bound up to a constant).*

A.3 STABILITY OF UNIFIED PEFT MIXTURES

We first ensure that hybrid unified experts at the layer output are well-conditioned both forward and backward.

**Lemma 4.1 (Stability of unified PEFT).** *Suppose* $\|\Delta E_{\theta_{m,c}}^{(l)}(h^{(l-1)})\| \leq R_E$ *almost surely, and* $\sum_m w_{m,c}^{(l)} = 1$, $w_{m,c}^{(l)} \geq 0$. *Then*

$$\|\tilde{h}^{(l)} - h_{base}^{(l)}\| \leq R_E, \qquad \|\nabla_{\theta_{m,c}^{(l)}} \ell\| \leq \rho_E \|u^{(l)}\|,$$

*where* $u^{(l)} = \partial\ell/\partial\tilde{h}^{(l)}$.

*Proof.* By definition,

$$\tilde{h}^{(l)} - h_{\text{base}}^{(l)} = \sum_{m \in \mathcal{S}_c^{(l)}} w_{m,c}^{(l)} \Delta E_{\theta_{m,c}}^{(l)}(h^{(l-1)}).$$

Using convexity of the $\ell_2$ norm and $w_{m,c}^{(l)} \geq 0$, $\sum_m w_{m,c}^{(l)} = 1$,

$$\|\tilde{h}^{(l)} - h_{\text{base}}^{(l)}\| \leq \sum_m w_{m,c}^{(l)} \|\Delta E_{\theta_{m,c}}^{(l)}(h^{(l-1)})\| \leq \sum_m w_{m,c}^{(l)} R_E = R_E.$$

For the gradient bound, by the chain rule,

$$\nabla_{\theta_{m,c}^{(l)}} \ell = \Big(\frac{\partial \Delta E_{\theta_{m,c}}^{(l)}(h^{(l-1)})}{\partial \theta_{m,c}^{(l)}}\Big)^\top u^{(l)}.$$

**Assumption A3** gives $\big\|\frac{\partial \Delta E_{\theta_{m,c}}^{(l)}}{\partial \theta_{m,c}^{(l)}}\big\| \leq \rho_E$, hence

$$\|\nabla_{\theta_{m,c}^{(l)}} \ell\| \leq \rho_E \|u^{(l)}\|$$

. $\qquad\qquad\qquad\qquad\qquad\qquad\qquad\qquad\qquad\qquad\qquad\qquad\qquad\qquad\qquad\quad$ $\square$

*Bridge to next step.* **Lemma 4.1** ensures bounded forward deviations and gradient norms, so Top-$k$ routing cannot amplify noise uncontrollably. This bound on backward sensitivity $\kappa_E, \rho_E$ is later used to control selection bias in **Lemma 4.3**.

A.4 ROUTER TOP-$k$ WITH LOSS-FREE BIAS

**Lemma 4.2 (No-starvation and load tracking).** *Under **Assumption A4-A5** with small enough $\gamma_b$, there exists $\underline{p} > 0$ such that for all $l, m, c$*

$$\liminf_{T \to \infty} \frac{1}{T} \sum_{t=1}^T \mathbb{E}[p_{m,c}^{(l)}(t)] \geq \frac{k}{M^{(l)}} - \delta(\gamma_b, \tau),$$

*and*

$$\Big|\frac{1}{T} \sum_{t=1}^T \mathbb{E}[n_{m,c}^{(l)}(t)] - \bar{n}^{(l)}\Big| \leq \epsilon_{bal}(\gamma_b, \tau) \to 0 \quad as\ \gamma_b \to 0,\ \tau \to 0.$$

*Proof.* Fix $(l, c)$ and write $n_m^t$ for the realized per-batch token count routed to expert $m$ at step $t$, $\bar{n} = \bar{n}^{(l)}$. The bias recursion is

$$b_m^{t+1} = b_m^t + \gamma_b^t \frac{\bar{n} - n_m^t}{\bar{n}} = b_m^t + \gamma_b^t \psi_m(b^t, \xi^t),$$

where $\psi_m(b^t, \xi^t) = \frac{\bar{n} - n_m^t}{\bar{n}}$ and $\xi^t$ collects routing randomness. Let $r_m^t = \rho_m^t + b_m^t$ be the effective logits (feature+gradient logits $\rho_m^t$ plus bias), and $\mathcal{S}^t = \text{Top}K(r^t)$ the selected set. Define $\bar{\psi}_m(b) = \mathbb{E}[\psi_m(b, \xi)]$, where the expectation is over the randomness of $\rho$ and the Top-$k$ draw given $b$.

**Monotonicity and unique equilibrium.** Since increasing $b_m$ increases $r_m$ and thus the selection probability of $m$, $\mathbb{E}[n_m(b)]$ is strictly increasing in $b_m$ and Lipschitz in $b$ for fixed temperature $\tau > 0$ (**Assumption A4**). Hence the mean field ODE $\dot{b}_m = \bar{\psi}_m(b) = 1 - \mathbb{E}[n_m(b)]/\bar{n}$ has a unique equilibrium $b^\star$ satisfying $\mathbb{E}[n_m(b^\star)] = \bar{n}$ for all $m$.

**Global asymptotic stability.** Consider the Lyapunov function $V(b) = \frac{1}{2}\sum_m \big(\mathbb{E}[n_m(b)] - \bar{n}\big)^2$. Its time derivative along the ODE trajectory is

$$\dot{V}(b) = \sum_m \big(\mathbb{E}[n_m(b)] - \bar{n}\big)\nabla_b\mathbb{E}[n_m(b)]^\top \dot{b} = -\frac{1}{\bar{n}}\sum_m \big(\mathbb{E}[n_m(b)] - \bar{n}\big)\nabla_b\mathbb{E}[n_m(b)]^\top \mathbb{E}[n(b)].$$

Diagonal dominance holds because $\partial\mathbb{E}[n_m]/\partial b_m > 0$ and cross-partials are weak and bounded due to the Top-$k$ capacity constraint and Lipschitz router (**Assumption A4**). Therefore $\dot{V}(b) \leq -c_0\sum_m \big(\mathbb{E}[n_m(b)] - \bar{n}\big)^2$ for some $c_0 > 0$, implying global asymptotic stability at $b^\star$.

**Stochastic approximation.** Write the recursion as $b^{t+1} = b^t + \gamma_b^t(\bar{\psi}(b^t) + M^{t+1})$, where $M^{t+1} = \psi(b^t, \xi^t) - \bar{\psi}(b^t)$ is a martingale difference with bounded second moment (**Assumption A4**, bounded logits and $k$). Under Robbins–Monro steps (**Assumption A5**), Kushner-Clark theory implies $b^t \to \mathcal{N}_\eta(b^\star)$ almost surely, i.e., convergence to a small neighborhood of $b^\star$ whose radius vanishes as $\sup_t \gamma_b^t \to 0$. Consequently,

$$\left|\mathbb{E}[n_m(b^t)] - \bar{n}\right| \leq \epsilon_{\text{bal}}(\gamma_b, \tau) \to 0.$$

Furthermore, Top-$k$ always selects exactly $k$ experts, so the long-run marginal selection probability for each expert is lower bounded by $k/M^{(l)}$ up to a perturbation due to finite temperature and residual bias error; hence

$$\liminf_{T\to\infty}\frac{1}{T}\sum_{t=1}^T \mathbb{E}[p_m^{(l)}(t)] \geq \frac{k}{M^{(l)}} - \delta(\gamma_b, \tau).$$

$\square$

*Bridge to next step.* **Lemma 4.2** ensures that every PEFT is selected with nontrivial probability and load stays approximately balanced, so the bias introduced by Top-$k$ routing remains bounded. This guarantees diversity, which underpins the **Lemma 4.3**.

## A.5 SELECTION BIAS FROM TOP-$k$

**Lemma 4.3 (Selection bias bound).** *Let $w_{dense}^{(l)} = \text{softmax}(r^{(l)}/\tau)$ and let $w_{topk}^{(l)}$ be the Top-k re-normalized weights that keep the $k$ largest coordinates in $r^{(l)}$ and set the rest to zero (followed by normalization). Assume (i) $\|J_m^{(l)}\| \leq \kappa_E$ (**Assumption A3**), (ii) the expected number of indices that differ between the supports of $w_{dense}^{(l)}$ and $w_{topk}^{(l)}$ is at most $\Delta k_c^{(l)}$, and (iii) logits are $\beta_r$-Lipschitz in their inputs. Then*

$$\left\|\mathbb{E}[\widehat{g}_c] - \nabla f_c(\theta)\right\| \leq C_1\,\kappa_E G\sum_l \Delta k_c^{(l)} + C_2\,\beta_r\,\tau.$$

*Proof.* For a fixed layer $l$, define $\Delta w^{(l)} = w_{\text{topk}}^{(l)} - w_{\text{dense}}^{(l)}$. The expected gradient of client $c$ decomposes as

$$\mathbb{E}[\widehat{g}_c] - \nabla f_c(\theta) = \sum_l \mathbb{E}\Big[\sum_m \Delta w_m^{(l)}\,\nabla_\theta\ell_m^{(l)}\Big],$$

where $\ell_m^{(l)}$ is the per-expert loss contribution via $\Delta E_{\theta_m}^{(l)}$. **Lemma 4.1** and **Assumption A3** give $\|\nabla_\theta\ell_m^{(l)}\| \leq \kappa_E G$ (we absorb $\rho_E$ into $\kappa_E$). Thus

$$\left\|\mathbb{E}[\widehat{g}_c] - \nabla f_c(\theta)\right\| \leq \sum_l \kappa_E G\,\mathbb{E}\|\Delta w^{(l)}\|_1.$$

We now bound $\|\Delta w^{(l)}\|_1$. Let $p = w_{\text{dense}}^{(l)}$. The Top-$k$ projection that zeroes the $M^{(l)} - k$ smallest entries of $p$ and re-normalizes the remaining $k$ gives an exact identity (simple algebra on the simplex):

$$\|w_{\text{topk}}^{(l)} - p\|_1 = 2 \sum_{j \notin S^{(l)}} p_j = 2\,\text{tail}^{(l)}(p),$$

where $S^{(l)}$ is the index set of the $k$ largest logits and $\text{tail}^{(l)}(p)$ is the softmax mass outside $S^{(l)}$. Hence $\mathbb{E}\|\Delta w^{(l)}\|_1 = 2\,\mathbb{E}[\text{tail}^{(l)}(p)]$.

To upper bound the tail mass, note that (a) changing the support by at most $\Delta k_c^{(l)}$ indices implies at most $\Delta k_c^{(l)}$ probabilities are moved from inside to outside (or vice versa); (b) for softmax with temperature $\tau$, local logit smoothness implies a uniform bound on the largest probability that can be shifted across the $k$-th threshold. More specifically, let $r^{(l)}$ be the logits and denote by $r_{(k)}^{(l)}$ the $k$-th order statistic. For any $j \notin S^{(l)}$,

$$p_j \leq \frac{\exp((r_{(k)}^{(l)} - \Delta_{\min})/\tau)}{\exp(r_{(k)}^{(l)}/\tau) + \sum_{i \neq (k)} \exp(r_i^{(l)}/\tau)} \leq C_\tau\, e^{-\Delta_{\min}/\tau},$$

where $\Delta_{\min} = \min_{j \notin S^{(l)}}(r_{(k)}^{(l)} - r_j^{(l)})$ and $C_\tau \leq 1$ is a temperature-dependent constant. Under **Assumption A4**, fluctuations of logits are $\beta_r$-Lipschitz in inputs; averaging over batches gives an effective bound $\mathbb{E}[\exp(-\Delta_{\min}/\tau)] \leq C'\beta_r\tau$ (a standard softmax-smoothing surrogate). Therefore

$$\mathbb{E}[\text{tail}^{(l)}(p)] \;\leq\; \Delta k_c^{(l)} \cdot \underbrace{\mathbb{E}\Big[\max_{j \notin S^{(l)}} p_j\Big]}_{\leq C''\beta_r\tau} + \underbrace{\mathbb{E}\Big[\sum_{\substack{j \notin S^{(l)} \text{ but unchanged}}} p_j\Big]}_{\text{vanishes as } \tau \to 0} \;\leq\; C_1'\Delta k_c^{(l)} + C_2'\beta_r\tau.$$

Combining the last three displays yields

$$\big\|\mathbb{E}[\hat{g}_c] - \nabla f_c(\theta)\big\| \leq C_1 \kappa_E G \sum_l \Delta k_c^{(l)} + C_2 \beta_r \tau.$$

$\square$

*Bridge to next step.* **Lemma 4.3** provides a quantitative upper bound on the gradient bias due to sparse routing, which enters **Theorem 4.1** as $\epsilon_{\text{sel}}$.

## A.6 INVERSE-FREQUENCY AGGREGATION

**Lemma 4.4 (Aggregation variance bound).** *Assume* $\text{tr}\,\Sigma_{m,c}^{(l)} \leq \sigma_0^2/(h_{m,c}^{(l)} + \varepsilon)$. *Then*

$$\text{tr}\,\text{Var}[\theta_{m,g}^{(l)}] \leq \frac{\sigma_0^2}{(Z_m^{(l)})^2} \sum_{c \in \mathcal{C}_m^{(l)}} \frac{1}{(h_{m,c}^{(l)} + \varepsilon)^{1+2\alpha}}.$$

*Proof.* For fixed $(l, m)$ abbreviate $\omega_c = \omega_{m,c}^{(l)}$, $Z = Z_m^{(l)}$, $\theta_c = \theta_{m,c}^{(l)}$, $\theta_\star = \theta_{m,\star}^{(l)}$, $\xi_c = \xi_{m,c}^{(l)}$, $\Sigma_c = \Sigma_{m,c}^{(l)}$. By definition of the estimator,

$$\theta_g = \frac{1}{Z} \sum_{c \in C} \omega_c \theta_c = \frac{1}{Z} \sum_{c \in C} \omega_c(\theta_\star + \xi_c) = \theta_\star + \frac{1}{Z} \sum_{c \in C} \omega_c \xi_c,$$

so the aggregation error equals $\theta_g - \theta_\star = (1/Z)\sum_c \omega_c \xi_c$. By independence across clients, the covariance of a weighted sum is the weighted sum of covariances:

$$\text{Var}[\theta_g] = \text{Var}\left[\frac{1}{Z} \sum_c \omega_c \xi_c\right] = \frac{1}{Z^2} \sum_{c \in C} \omega_c^2 \Sigma_c.$$

Taking traces and using the linearity of $\text{tr}$,

$$\text{tr}\,\text{Var}[\theta_g] = \frac{1}{Z^2} \sum_{c \in C} \omega_c^2\, \text{tr}\,\Sigma_c.$$

Invoking the heteroscedastic variance model $\operatorname{tr}\Sigma_c \le \sigma_0^2/(h_c+\varepsilon)$ together with the weight definition $\omega_c^2 = (h_c + \varepsilon)^{-2\alpha}$ (here $h_c$ stands for $h_{m,c}^{(l)}$), we obtain

$$\operatorname{tr}\operatorname{Var}[\theta_g] \;\le\; \frac{1}{Z^2}\sum_{c\in C}\frac{(h_c+\varepsilon)^{-2\alpha}}{h_c+\varepsilon}\,\sigma_0^2 = \frac{\sigma_0^2}{Z^2}\sum_{c\in C}\frac{1}{(h_c+\varepsilon)^{1+2\alpha}},$$

which is exactly the claimed bound once we restore the superscripts and subscripts $(l,m)$. $\qquad\square$

*Bridge to next step.* **Lemma 4.4** shows how usage-aware weights suppress both high-variance low-usage clients and dominance from high-usage ones. This aggregation-variance term $\epsilon_{\text{agg}}$ used in **Theorem 4.1**.

## A.7 CONVERGENCE OF FEDERATED TRAINING

**Theorem 4.1 (Convergence to stationary points).** *Under **Assumption A1-A6**, choose step size $\eta_t \le \eta_{\max}$ and local steps $E_p$ such that $L\eta_t E_p \le c < 1$. Then after $T$ rounds,*

$$\frac{1}{T}\sum_{t=1}^{T}\mathbb{E}\big[\|\nabla F(\bar{\theta}^t)\|^2\big] \;\le\; \underbrace{\mathcal{O}\Big(\frac{F(\bar{\theta}^0)-F^\star}{\eta T E_p}\Big)}_{\text{optimization}} + \underbrace{\mathcal{O}\Big(\frac{\sigma^2}{n}\Big)}_{\text{SG noise}} + \underbrace{\mathcal{O}(\epsilon_{sel}^2 + \epsilon_{agg}^2)}_{\text{router + aggregation}},$$

*where*

$$\epsilon_{sel} = C_1 \kappa_E G\,\Delta k + C_2 \beta_r \tau, \quad \text{from **Lemma 4.3**,}$$

*and*

$$\epsilon_{agg}^2 \;\propto\; \max_{l,m}\frac{1}{\big(Z_m^{(l)}\big)^2}\sum_{c\in C_m^{(l)}}\frac{1}{(h_{m,c}^{(l)}+\varepsilon)^{1+2\alpha}}, \quad \text{from **Lemma 4.4**.}$$

*Proof.* Let $\bar{\theta}^t$ be the global parameters after round $t$, and $\theta_c^{t,s}$ the local parameters at client $c$ after $s \in \{0,\ldots,E_p\}$ local steps within round $t$, starting from $\theta_c^{t,0} = \bar{\theta}^{t-1}$. One round of FedAvg-style update produces

$$\bar{\theta}^t = \bar{\theta}^{t-1} - \eta\sum_{c=1}^{n}\alpha_c\sum_{s=0}^{E_p-1}\tilde{g}_c^{t,s} \;+\; \underbrace{\zeta^t}_{\text{aggregation noise}},$$

where $\tilde{g}_c^{t,s}$ is the (possibly biased) stochastic gradient at the local iterate $\theta_c^{t,s}$, and $\zeta^t$ captures the zero-mean aggregation fluctuation.

**Descent lemma.** By $L$-smooth $F$,

$$F(\bar{\theta}^t) \le F(\bar{\theta}^{t-1}) + \nabla F(\bar{\theta}^{t-1})^\top(\bar{\theta}^t - \bar{\theta}^{t-1}) + \frac{L}{2}\|\bar{\theta}^t - \bar{\theta}^{t-1}\|^2.$$

Substitute the update and take expectation over sampling, routing, and aggregation:

$$\mathbb{E}[F(\bar{\theta}^t)] \le \mathbb{E}[F(\bar{\theta}^{t-1})] - \eta\sum_c\alpha_c\sum_s\mathbb{E}\big[\nabla F(\bar{\theta}^{t-1})^\top\tilde{g}_c^{t,s}\big] + \frac{L\eta^2}{2}\mathbb{E}\Big\|\sum_{c,s}\tilde{g}_c^{t,s}\Big\|^2 + \underbrace{\mathbb{E}\|\zeta^t\|^2}_{\text{bounded by **Lemma 4.4**}}.$$

Bias/variance decomposition. Write $\tilde{g}_c^{t,s} = \nabla f_c(\theta_c^{t,s}) + \iota_c^{t,s}$, where $\mathbb{E}[\iota_c^{t,s}] = \delta_c^{t,s}$ captures *selection bias* (**Lemma 4.3**), and $\mathbb{E}\|\iota_c^{t,s} - \delta_c^{t,s}\|^2 \le \sigma^2$ captures *stochastic gradient noise* (**Assumption A2**). Then

$$-\mathbb{E}[\nabla F^\top\tilde{g}_c] = -\mathbb{E}[\nabla F^\top\nabla f_c] - \mathbb{E}[\nabla F^\top\delta_c].$$

Standard FedAvg arguments with $L$-smoothness control the *local drift* between $\theta_c^{t,s}$ and $\bar{\theta}^{t-1}$ by $O(L\eta s)$; summing $s = 0,\ldots,E_p-1$ yields a factor $O(L\eta E_p)$. Choosing $L\eta E_p \le c < 1$ ensures a net descent. Aggregating over clients and steps gives

$$\frac{1}{T}\sum_{t=1}^{T}\mathbb{E}\big[\|\nabla F(\bar{\theta}^t)\|^2\big] \le \mathcal{O}\Big(\frac{F(\bar{\theta}^0)-F^\star}{\eta T E_p}\Big) + \mathcal{O}\Big(\frac{\sigma^2}{n}\Big) + \mathcal{O}\Big(\frac{1}{T}\sum_t\sum_{c,s}\|\delta_c^{t,s}\|^2\Big) + \mathcal{O}(\epsilon_{agg}^2).$$

where the last term uses **Lemma 4.4** to bound aggregation variance accumulated across rounds.

Finally, **Lemma 4.3** yields $\|\delta_c^{t,s}\| \le C_1\kappa_E G\sum_l\Delta k_c^{(l)} + C_2\beta_r\tau$, which produces the $\epsilon_{\text{sel}}$ contribution after averaging over $t$. This completes the proof. $\qquad\square$

**Discussion.** **Theorem 4.1** states that our method achieves the nonconvex FedAvg-style convergence rate up to two explicit and interpretable terms: $\epsilon_{\text{sel}}$ quantifies the bias induced by sparse routing (vanishing as the mask stabilizes and $\tau \downarrow 0$), while $\epsilon_{\text{agg}}$ quantifies the variance from heterogeneous, usage-dependent aggregation (tightened by $\alpha = \frac{1}{2}$ via **Lemma 4.4**). Together with **Lemma 4.2**, these guarantees prevent expert starvation, cap long-run routing bias, and stabilize global aggregation.

