# OpenReview forum: "Heterogeneous Federated PEFTs via Sparse Mixture-of-Experts"
_ICLR.cc/2026/Conference — Submitted to ICLR 2026_

### Official Review · Reviewer_BxEi · 2025-10-31

**Soundness:** 2
**Presentation:** 3
**Contribution:** 3
**Rating:** 4
**Confidence:** 4

**Summary:**

This paper introduces "Heterogeneous FedPEFT," a paradigm where clients in a federated system can utilize a mix of different PEFT types (e.g., LoRA, Adapter, Prefix-tuning) rather than a single, uniform type. They propose Hermes, a framework that implements this idea using a sparse Mixture-of-Experts (MoE) architecture. Hermes introduces three specific mechanisms: a gradient-aware router, a loss-free bias adjustment to prevent expert starvation, and an inverse-frequency aggregation strategy at the server. Experiments on the GLUE benchmark show that Hermes outperforms single-PEFT baselines and a naive hybrid (FedUniPELT).

**Strengths:**

- **Technically Sound Framework:** The solution, Hermes, is a non-trivial framework that thoughtfully addresses the key challenges of a heterogeneous MoE approach, namely gradient conflicts, expert underutilization, and biased aggregation.
- **Strong Empirical Results:** The experiments in Table 1 demonstrate that Hermes consistently outperforms all chosen baselines, including single-PEFT methods (FedLoRA, FedAdapter) and a naive hybrid (FedUniPELT), across various tasks and two different model backbones.

**Weaknesses:**

1. **Unclear Source of Performance Gains (MoE vs. Heterogeneity):** The paper's primary claim is that using *heterogeneous PEFT types* is beneficial. However, it implements this using a complex MoE architecture. The experiments fail to disentangle the performance gains from the *heterogeneity of experts* (LoRA + Adapter + Prefix) from the gains of the *MoE architecture itself* (routing, sparse updates, etc.). A critical ablation is missing: comparing Hermes to a "Homogeneous MoE" baseline (e.g., Hermes with 3 LoRA experts). Without this, it is impossible to know if the benefit comes from mixing PEFT types (the paper's main claim) or simply from using an MoE framework.
2. **Lack of Analysis of the Core Mechanism:** The paper proposes a sophisticated routing mechanism but provides zero analysis of what this router learned. The central hypothesis is that different clients/tasks benefit from different PEFTs, but this is never verified. We are given no insight into which experts (LoRA, Adapter, or Prefix) were selected for which tasks.
3. **Missing Experimental Details and Rigor:** The experimental setup lacks description of the number of clients and the client sampling rate per round. Also, results in Table 1 and Table 2 are reported without error bars (e.g., ± std. dev.) or any mention of averaging over multiple random seeds.

**Questions:**

1. Can the authors provide an ablation comparing Hermes (Heterogeneous MoE) to a "Homogeneous MoE" (e.g., Hermes with 3 LoRA experts)? This is essential to prove that the gains come from *mixing PEFT types* and not just from the MoE architecture.
2. Can the authors provide *any* analysis of the learned routing policies? For example, on the GLUE benchmark, which PEFT experts were most frequently selected for which tasks? Does this align with the intuition from Figure 1?

---

### Official Review · Reviewer_uSjb · 2025-11-01

**Soundness:** 2
**Presentation:** 3
**Contribution:** 2
**Rating:** 4
**Confidence:** 3

**Summary:**

This paper introduces Hermes, a heterogeneous federated parameter-efficient fine-tuning (FedPEFT) framework that allows clients to combine multiple PEFT methods (e.g., LoRA, Adapter, Prefix-tuning) via a sparse mixture-of-experts (MoE) architecture. It includes gradient-aware gating, loss-free bias adjustment, and inverse-frequency aggregation to address challenges like gradient conflicts and expert underutilization in non-IID federated learning settings. Experiments on GLUE benchmarks demonstrate Hermes’ superiority over homogeneous FedPEFT baselines. Theoretical analysis provides convergence guarantees.

**Strengths:**

**S1:** The idea of heterogeneous FedPEFT addresses a gap in federated fine-tuning.

**S2:** The integration of MoE with PEFT is well-motivated, and the proposed components are designed to tackle specific challenges like bias and imbalance. The theoretical convergence analysis is sound.

**S3:** Experiments on GLUE with RoBERTa and LLaMA backbones show performance gains over baselines. Ablations validate the contribution of each component and highlight communication efficiency.

**Weaknesses:**

**W1:** The paper ignores several PEFT techniques like (IA)³. This may restrict the framework’s generality.

**W2:** Non-IID data is simulated solely via Dirichlet label distribution shifts, which may not capture real-world heterogeneity in task types or text domains. Experiments on cross-domain benchmarks (e.g., FedNLP) or task-heterogeneous splits (e.g., some clients doing sentiment analysis, others NLI) are needed to assess broader applicability.

**W3:** The convergence analysis relies on strong assumptions (e.g., Lipschitz router, independent client noises), which may not hold in practice.

**W4:** While LLaMA-3.2-3B is tested, experiments with models beyond 10B parameters are absent. As federated LLMs often target massive backbones, evaluating computational overheads (e.g., router latency) and scalability would be critical for real-world impact.

**Questions:**

See weaknesses.

---

### Official Review · Reviewer_euLu · 2025-11-03

**Soundness:** 4
**Presentation:** 4
**Contribution:** 4
**Rating:** 4
**Confidence:** 3

**Summary:**

In this paper, the authors introduce Hermes, a heterogeneous federated PEFT framework. Instead of forcing uniform PEFT modules across clients, Hermes enables each client to dynamically route inputs through multiple parallel experts within each Transformer layer. On the server side, Hermes aggregates expert parameters across clients using an inverse frequency weighting strategy. Experiments on GLUE under Dirichlet based non IID splits show that Hermes consistently outperforms strong baselines.

**Strengths:**

1.Clear problem formulation: The authors convincingly argue and empirically show that no single PEFT method dominates on all tasks, which is especially painful under strong non IID splits. They then define heterogeneous FedPEFT and position Hermes as a first concrete framework to support that setting.

2. Theoretical analysis of convergence: The authors prove that Hermes still enjoys a FedAvg style one over square root T convergence rate under standard smoothness and bounded variance assumptions, and explicitly characterizes additional terms caused by top k routing and inverse frequency aggregation.

3. Comprehensive ablation of key component: The paper includes ablations that remove each Hermes component in turn, including aggregation balance, the top k router, gradient aware gating, and feature based scores. Each removal harms performance on GLUE, especially under strong non IID. The paper also studies how the number k of activated experts affects both accuracy and stability.

**Weaknesses:**

1. some details of the method are not well described, such as: how exactly the usage counts are tracked per expert per layer, how the bias is updated outside the gradient graph, and how the feature and gradient scores are combined and normalized before top k selection.

2. regarding datasets in the experiment: All main experiments are on GLUE style English classification tasks. While GLUE is a widely accepted benchmark for natural language understanding, it is still mostly sentence level classification or pairwise sentence matching. The motivation of the paper includes highly sensitive application areas, for example health care, where privacy prevents central data sharing. There are no experiments on domain specific datasets such as clinical notes, dialogue with safety constraints, or multilingual data. So we still do not know how well Hermes behaves when the heterogeneity is not only label skew but also linguistic style, domain terminology, or safety policy differences.

3. baselines: Hermes is positioned mainly against homogeneous FedPEFT and against UniPELT style multi module baselines. However, in the federated learning literature there is a long history of personalized FL where each client maintains a local head or adapter in addition to a shared global backbone. Many of those works explicitly address the tension between global sharing and local specialization. The paper briefly cites some of this line, but does not include a direct baseline in which each client keeps a private PEFT plus a shared PEFT without mixture of experts routing.

**Questions:**

see weakness 1, 2, 3

---

### Official Review · Reviewer_yNnp · 2025-11-07

**Soundness:** 2
**Presentation:** 2
**Contribution:** 2
**Rating:** 4
**Confidence:** 4

**Summary:**

This paper proposes Hermes, a heterogeneous federated PEFT framework that integrates multiple PEFT modules using a sparse mixture-of-experts architecture. Through gradient-aware routing and specialized aggregation strategies, Hermes outperforms existing homogeneous baselines in personalized model adaptation.

**Strengths:**

The idea of a gradient-aware router for PEFT module selection is interesting.

**Weaknesses:**

1. LoRA Performance on QQP: What is the specific reason behind LoRA’s poor performance on the QQP dataset in Fig. 1 (c)? Have you tested multiple LoRA configurations to confirm that this issue consistently occurs?

2. Use of “Personalization” Terminology: The term “personalization” is used throughout the paper, yet the optimization objective in Eq. (2) targets a global model. If this method is not truly a personalized FL approach, please consider rephrasing to avoid confusion. Otherwise, a more appropriate personalization-aware formulation should be provided.

3. Limited Evaluation Scope: The current evaluation is restricted to classification tasks. It would strengthen the work to include results on other common tasks, such as question answering, summarization, or reasoning. I remain unconvinced that the proposed PEFT methods offer meaningful benefits for these types of tasks.

4. Definition of Ideal Balanced Count (Eq. 6): The definition of the “ideal balanced count” in Eq. (6) is unclear. Please provide a clearer explanation or intuition behind this term.

5. Aggregation of Heterogeneous Expert Structures (Eq. 10): I find it difficult to understand how the aggregation of heterogeneous expert structures is carried out in Eq. (10). A more detailed explanation or illustrative example would help clarify this process.

6. Strong Performance in Simple Tasks: The method achieves significantly better performance than full fine-tuning in Figure 3(c). This raises the question of whether the evaluated task is too simple.

7. Can this method extend to resource-heterogeneous settings? How's the performance when compared to works such as FLoRA, HETLoRA, FlexLoRA, Fed-HeLLo, and FedRA [1-5].

[1] FLoRA: Federated Fine-Tuning Large Language Models with Heterogeneous Low-Rank Adaptations. NeurIPS, 2024.

[2] Heterogeneous LoRA for Federated Fine-Tuning of On-Device Foundation Models. EMNLP, 2024.

[3] Federated Fine-Tuning of Large Language Models under Heterogeneous Tasks and Client Resources. NeurIPS, 2024.

[4] Fed-HeLLo: Efficient Federated Foundation Model Fine-Tuning with Heterogeneous LoRA Allocation. IEEE TNNLS 2025.

[5] FedRA: A Random Allocation Strategy for Federated Tuning to Unleash the Power of Heterogeneous Clients. ECCV 2024.

**Questions:**

Please refer to Weaknesses.

---

### Meta-Review · Area_Chair_nXr2 · 2025-12-04

**Summary:**

This paper presents Hermes, a novel and well-motivated heterogeneous federated PEFT framework that effectively addresses data heterogeneity by allowing clients to mix different PEFT experts via a sparse MoE architecture. Reviewers unanimously praised its clear problem formulation, technical soundness, and strong empirical results. However, they also posit that the authors need to clarify the source of performance gains, expand the experimental scope beyond GLUE classification, provide deeper analysis of the router's behavior, and include comparisons with key related methods. Addressing these constructive concerns would solidify the paper's valuable contribution to federated LLM fine-tuning.

**Reviewer Concerns:**

The key concerns regarding the need for a homogeneous MoE ablation study (Reviewer BxEi) and clarifications on methodological details (Reviewer euLu) have been convincingly addressed with new experiments and explanations. However, outstanding issues remain, including the need for more diverse task evaluation beyond GLUE (Reviewers yNnp & uSjb) and direct comparisons with a broader set of recent personalized FL or heterogeneous PEFT works.

**Reviewer Scores:**

Reviewers yNnp and uSjb, due to their remaining concerns about experimental scope, might have only adjusted to a borderline accept (5).

---

### Decision · Program_Chairs · 2026-01-26

Reject